# A Comparative Study of Initial Cost Recuperation Period of Plug-In Series Hybrid Electric Two-Wheel Vehicles in Southeast Asian Countries

**Woongchul Choi [1],* and Seokho Yun [2]**

1   Department of Automotive Engineering, Kookmin University, Seoul 02707, Korea
2   Department of Security Enhanced Smart Electric Vehicle, Kookmin University, Seoul 02707, Korea;
    seokho@kookmin.ac.kr
*   Correspondence: danchoi@kookmin.ac.kr

**Abstract:** While pollutant emissions from vehicles are under heavy scrutiny all around the world, small two-wheelers have not been under strict regulations until recently. Especially in the region of Southeast Asia, a tremendous number of old, in-house rebuilt and outdated two-wheelers are in operation and, as a result, pollutant emission problems are one of the most serious concerns of the communities. Since electric grid systems for consistent and stable supply of electricity are not there yet, thus plug-in series hybrid two-wheel vehicles have attracted much attention and are thought to be a meaningful solution for many people in the region. In the current study, an energy simulation tool has been developed to compare the ownership cost of an internal combustion engine (ICE)-based two-wheeler and that of a plug-in series hybrid electric scooter. To estimate annual energy cost (sum of gasoline and electricity cost), gasoline prices and household electricity rates in major Southeast Asian countries were collected. In addition, the nominal initial vehicle prices of ICE-based scooters and those of plug-in series hybrid electric two-wheel vehicles were gathered to estimate the time for the recovery of the initial investment.

**Keywords:** hybrid electric two-wheel vehicle; hybrid electric scooter; battery; driving cycle; investment cost recovery

## 1. Introduction

As a part of ongoing efforts to reduce the amount of pollutant emissions and $CO_2$ production from ICE-based conventional vehicles, more detailed and stricter regulations are being imposed [1]. Beginning in 1999, two-wheel vehicles have been under regulation in European countries as specified in the Euro 1 emission control standard. Since then, the restrictions have been strengthened from Euro 1, 2, 3, 4, and finally up to Euro 5 in January 2020 [2]. The emission limits for two-wheel vehicles in Euro 5 are 1.0 g/km of CO, 1.0 g/km of HC, 0.06 g/km of $NO_x$, and not more than 0.0045 g/km of particulate matter (PM) [3]. Furthermore, starting with Euro 5, two-wheel vehicles must have an on-board diagnostic as well [2–4]. In many Southeast Asian countries, one of the most popular transportation methods is two-wheelers [5]. Among those countries, Indonesia has the highest number of two-wheel vehicles registered, as high as about one two-wheel vehicle per two people [6]. The Philippines has a relatively lower number of registered two-wheelers, but not by much: about one for every three people [7]. The real problem with these countries is the fact that most ICE-based conventional vehicles are loosely controlled in terms of emission regulation [8]. As a result, the air pollution caused by driving two-wheel vehicles is intensifying, not only because of the huge number of two-wheelers, but also due to the failure of effective regulation control [9–11]. In this paper, rather simple and straightforward

energy consumption models were developed, one for an ICE-based two-wheel vehicle and the other for a plug-in series hybrid electric two-wheel vehicle. With the simulation models constructed using Matlab Simulink, comparative studies to investigate the recuperation period of the initial investment for the plug-in series hybrid electric two-wheelers were conducted. For the current study, four major Southeast Asian countries, namely Thailand, the Philippines, Indonesia, and Vietnam, were carefully evaluated considering the initial cost and the estimated total annual energy costs. The results from the study are detailed in this paper.

## 2. Vehicle Specifications and Structure

A typical two-wheel vehicle specification was selected for the current comparative study. For an ICE-based conventional two-wheeler, a 125 cubic centimeter (cc) gasoline engine was selected as the main powertrain. For a plug-in series hybrid electric two-wheeler, a 3 kW brushless direct current (BLDC) motor was picked as the main powertrain with a lithium battery pack as energy storage, and a 125 cc ICE was installed as a generator system for simplicity. The component layouts of the ICE-based two-wheeler and the plug-in series hybrid electric two-wheeler are illustrated in Figure 1.

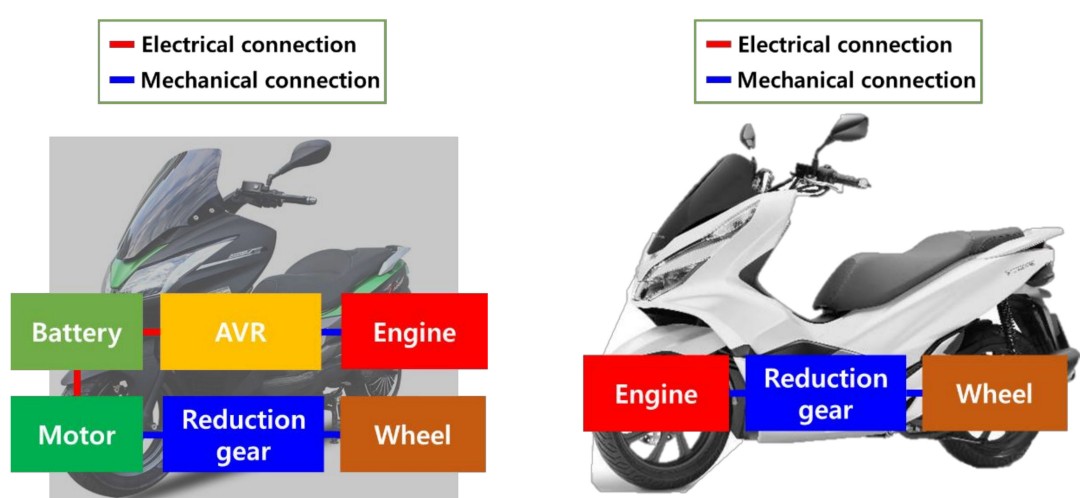

(**a**) Component layout of plug-in series hybrid electric two-wheel vehicle.　(**b**) Component layout of ICE two-wheel vehicle.

**Figure 1.** Component layouts of the two-wheel vehicles.

## 3. Current Status of Electricity Distribution in Major Countries in Southeast Asia

Major countries in Southeast Asia are suffering from a significant shortage of electricity [12]. According to the International Energy Agency, France, about 65 million people, or 10% of the Southeast Asia population, suffer from a situation where they cannot enjoy the benefits of electricity [13]. The main reason for the power shortage in Southeast Asia is the surging demand for energy due to the rapidly changing economy [14]. Thailand has a high electricity supply rate. The cities of northern Thailand have about 100% electricity penetration and even the rural areas have about 99.8%. However, the southern area of Thailand still has problems with chronic power shortages [15]. The Philippines has a quite different story. Some areas of the Philippines have a significantly lower electricity distribution rate at 74.1% and suffer from power shortages [16]. The average electricity distribution rate in Indonesia's 21 provinces is approaching 90%, but in some provinces the electricity penetration rates are still low [17]. In Vietnam, the electricity penetration rate is relatively high [18]. However, overall electricity consumption grows even higher as Vietnamese industry is expanding at full speed, therefore limiting the amount of available electricity for individual households. More importantly, the main portion of the electricity in Vietnam is generated by coal-based power plants, thus having a great influence on air

pollution [19]. Due to lack of coal supply and excessive pollutant emissions from power generation facilities, it is fully expected to be regulated down to produce a limited amount of electricity [20].

Due to the aforementioned situations of the electricity supply and infrastructure in Southeast Asian countries, plug-in series hybrid electric two-wheelers are more favorable than pure electric two-wheelers. Therefore, in this study, comparison of the ICE two-wheeler and the plug-in series hybrid electric two-wheeler was carefully conducted. Additionally, in order to understand the optimal size of battery packs, various scenarios of charge depleting (CD) operation range (electric consumption only), and charge sustaining (CS) operation range (electric consumption and generation) were evaluated with the estimation of annual energy costs. A fully electric two-wheeler could give a lower overall operating cost but was beyond the scope of the current study as the research focus was on the plug-in hybrid electric two-wheeler under the influence of the rather unstable power grid situation.

## 4. Simulation Environment

As a representative driving pattern of two-wheeled vehicles, the New York City Cycle (NYCC) test was selected as shown in Figure 2, since it reflects downtown style accelerations and decelerations mainly in urban areas. As can be seen in the NYCC results, the driving distance of 1.889 km took 598 s for a single run, which is much less than a typical daily driving distance. Therefore, a reasonable estimated daily driving distance needed to be selected. With the lack of clear information from Southeast Asian countries, a driving distance of 39.2 km, which is the average daily driving distance of Korea in 2018, was used for the current study, resulting in 21 iterations of NYCC tests to fulfil the assumed daily driving distance [21].

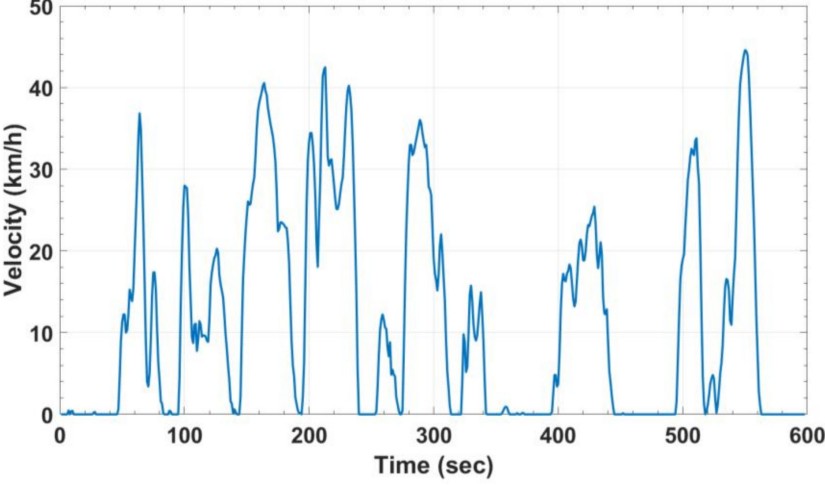

**Figure 2.** New York City Cycle.

An in-house forward simulation model was developed in the Matlab environment as illustrated in Figure 3. A system of a vehicle control unit, a BLDC motor for traction, an auxiliary engine for electricity generation, and a lithium ion battery for energy storage was configured to complete the plug-in series hybrid electric two-wheel vehicle model. With appropriate control strategy implemented, a variety of charge depleting (CD) and charge sustaining (CS) scenarios were evaluated for comparative studies among four counties, Thailand, the Philippines, Indonesia, and Vietnam. Prior to the comparative evaluation of various driving scenarios, model verifications were carried out against the well-known energy analysis tool Autonomie® from Argonne National Lab. The consumption of the fuel and the electricity from the same vehicle configuration were carefully monitored and compared between the in-house model and Autonomie. During this validation process, the motor efficiency map (Figure 4) and the engine efficiency map (Figure 5) were utilized also; the simple energy flow from the battery was used in both models. Since the main purpose of the current research was to evaluate the merit of two different vehicle configurations, overall energy consumption was closely monitored.

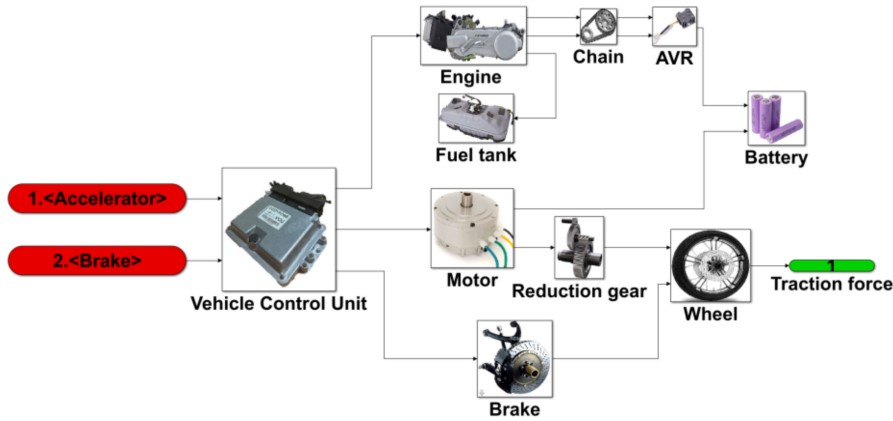

**Figure 3.** Schematic of energy analysis model.

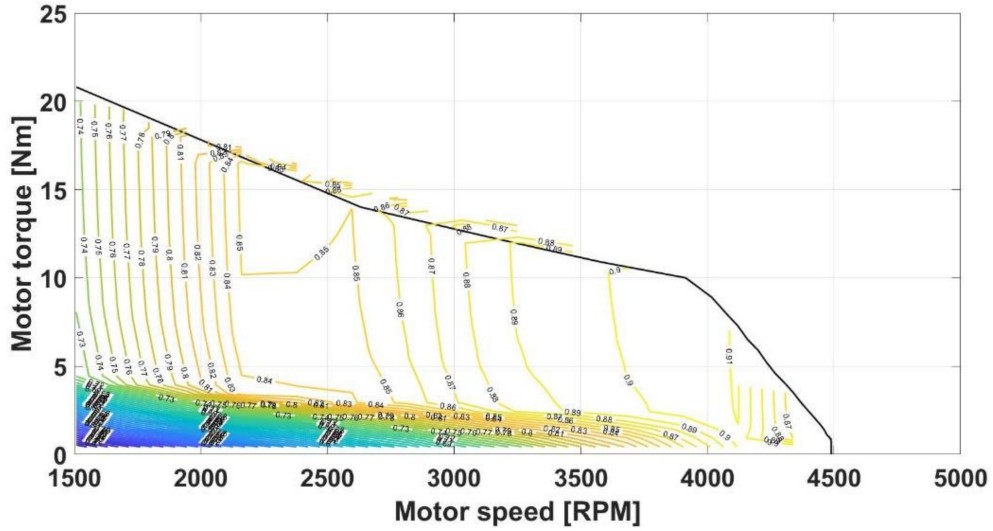

**Figure 4.** Efficiency map of the 3.0 kW brushless direct current (BLDC) motor used in the simulation.

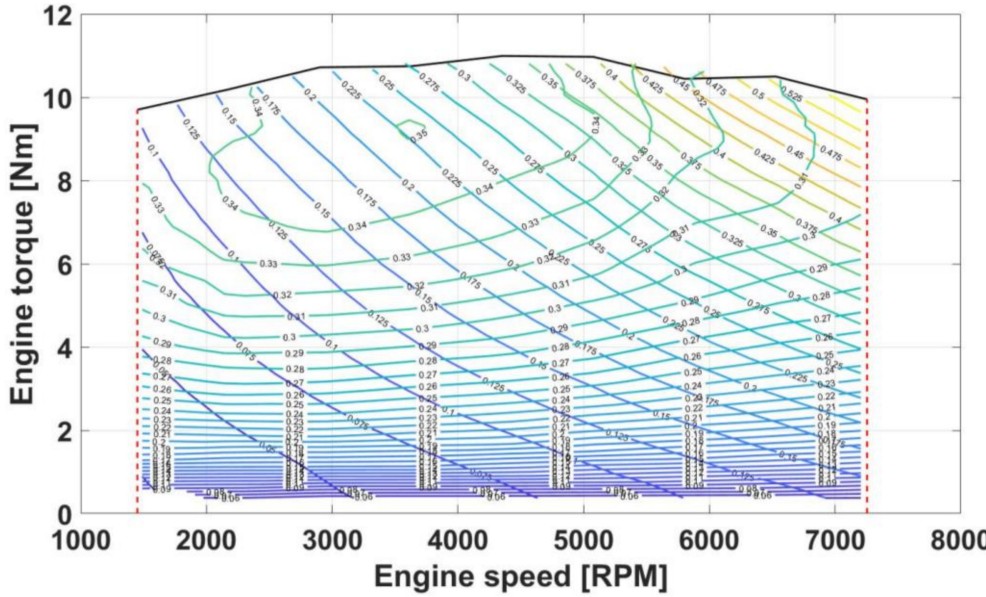

**Figure 5.** Efficiency and fuel consumption map of the 125 cc internal combustion engine (ICE) used in the simulation.

## 5. Simulations

In order to organize various simulation results, case numbers were assigned to each country, namely, Case 1 and 2 to Thailand, Case 3 and 4 to the Philippines, Case 5 and 6 to Indonesia, and finally Case 7 and 8 to Vietnam. Case 1, Case 3, Case 5, and Case 7 contain the results of the simulations for an ICE two-wheeler with a daily driving distance of 39.2 km/day. The annual fuel costs for each country, as mentioned above, were listed for comparative study. Case 2, Case 4, Case 6, and Case 8 include the results of the simulations for a plug-in series hybrid two-wheeler with the same driving distance. For each of these cases, different combinations of CD/CS scenarios were simulated with various sizes of batteries for further assessment of the possible optimal configuration of the CD/CS driving strategy. Just as before, the annual fuel costs for each country were listed for amenable comparative analysis.

The initial purchase cost of a typically available ICE two-wheel vehicle was obtained through data survey at the Alibaba internet shopping mall and the cost of a plug-in series hybrid electric two-wheel vehicle was estimated by simply adding the motor, controller, and battery cost. In this estimation, cost of the engineering development was not considered in order to compare the products in mass production. The total annual cost was estimated simply by adding the initial two-wheel vehicle price to the annual fuel cost for all cases. Depreciation amounts for different vehicle configurations were not considered for simplicity and we assumed no depreciation. In order to establish an average battery capacity for the plug-in series hybrid electric two-wheeler, reference data from the Ministry of Environment of Korea were adopted [22]. Typical pure electric two-wheelers were found to be equipped with 2.5 kWh of battery capacity. Based on this reference, three different configurations of 1.5 kWh, 2.0 kWh, and 2.5 kWh battery sizes were investigated.

For the best possible comparative merit analysis, Korean currency (won-to-US dollar or USD) comparisons were tabulated and plotted respectively for each case. Gasoline prices and electricity rates for regular households in Thailand, the Philippines, Indonesia, and Vietnam were surveyed accordingly and then converted into US dollars based on the average exchange rates posted for the period of the past year as provided in Table 1 [23,24] (1132.31 won = 1 USD).

**Table 1.** Major Southeast Asian countries' gasoline and electricity prices.

|  | Thailand | The Philippines | Indonesia | Vietnam |
|---|---|---|---|---|
| Gasoline price (USD/L) | 1.21 | 1.05 | 0.74 | 0.92 |
| Electricity price (USD/kWh) | 0.11 | 0.19 | 0.11 | 0.09 |

From the initial observation simply based on the fuel and electricity costs, it is favorable to use electricity as much as possible for all countries. However, as further investigations were carried out, we found each country faced challenges because of unreliable electrical supply due to unstable power grids and infrastructure, especially at the end of distribution line, in other words, right at the point of connection to residential houses.

Notably, in the Philippines, the cost of electricity is quite high compared to the other three countries. This may mean that the dependency on electric power may be less favorable. In the case of Indonesia, the cost of gasoline is a lot cheaper as Indonesia is one of the major petroleum production countries. As for Thailand and Vietnam, power shortages are known to be frequent due to the lack of well-established infrastructure for the final end distribution of the electricity to regular households, while the cost of electricity is relatively low.

## 6. Simulation Results and Analysis

In the current paper, results and analysis for Thailand and the Philippines are reported first.

### 6.1. Thailand

In Thailand, the rationale for using more electric energy than gasoline seems strong and reasonable based on the cost structure. However, in reality, gasoline was considered to be more dependable regardless of the higher cost compared to that of electricity. Consumer expectations for the reliability of transportation are severely reduced due to the lack of stable electricity infrastructure down at the household level. With this complication, it is quite difficult to come up with an obvious recommendation based on the cost comparison only. However, as a first effort to provide a rationale for the use of $CO_2$ and pollutant emission-free energy, adoption of a higher dependency on electrically powered vehicles is recommended with a recuperation period of 1.2 years. Detailed discussion is provided in the following.

The results from the configuration of Case 1 represent the annual fuel cost incurred if the driving distance of 39.2 km is solely achieved by ICE in Thailand. With the gasoline price at 1.2 USD/L, a daily fuel cost of 0.92 USD is estimated for the ICE two-wheel vehicle, thus resulting in an annual fuel cost of 335.24 USD.

Under the system configuration of Case 2, a number of simulations for various sizes of battery (1.5 kWh, 2.0 kWh, and 2.5 kWh) were carried out. The annual energy cost for each battery is shown in Tables 2–4 with the CD range changing while the CS range is fixed. More specifically, with the minimum state of charge (SOC) of the lithium ion battery set at 20%, the initial battery SOC was decreased by 10%, from 100% down to 70%. Additionally, the CS range was fixed at 30% to 20% to estimate the annual energy cost for various CD ranges. For example, the scenario of 90~20% of CD range and 30~20% of CS range means that the electricity from the battery with SOC 90% down to 20% is used first and then the ICE generator kicks in to produce electricity until the SOC reaches 30%. Once the SOC reaches 30%, then the ICE stops and the traction motor simply uses the electricity again from 30% SOC down to 20% SOC. This process repeats until the total driving distance of 39.2 km/day is achieved.

**Table 2.** Lithium ion battery capacity 1.5 kWh result (charge sustaining (CS) range fixed).

| Initial SOC (%) | CD Range (%) | CS Range (%) | Annual Gasoline Cost (USD) | Annual Electricity Cost (USD) | Annual Energy Cost (USD) |
|---|---|---|---|---|---|
| 100 | 100~20 | 30~20 | 184.35 | 53.00 | 237.35 |
| 90 | 90~20 | 30~20 | 195.25 | 50.84 | 246.09 |
| 80 | 80~20 | 30~20 | 236.14 | 39.42 | 275.56 |
| 70 | 70~20 | 30~20 | 245.80 | 37.24 | 283.05 |

**Table 3.** Li-ion batt capacity 2.0 kWh result (CS range fixed).

| Initial SOC (%) | CD Range (%) | CS Range (%) | Annual Gasoline Cost (USD) | Annual Electricity Cost (USD) | Annual Energy Cost (USD) |
|---|---|---|---|---|---|
| 100 | 100~20 | 30~20 | 120.30 | 68.30 | 188.60 |
| 90 | 90~20 | 30~20 | 142.84 | 62.94 | 205.78 |
| 80 | 80~20 | 30~20 | 178.13 | 53.58 | 231.71 |
| 70 | 70~20 | 30~20 | 219.60 | 42.26 | 261.86 |

**Table 4.** Li-ion batt capacity 2.5 kWh result (CS range fixed).

| Initial SOC (%) | CD Range (%) | CS Range (%) | Annual Gasoline Cost (USD) | Annual Electricity Cost (USD) | Annual Energy Cost (USD) |
|---|---|---|---|---|---|
| 100 | 100~20 | 30~20 | 56.47 | 84.83 | 141.29 |
| 90 | 90~20 | 30~20 | 74.15 | 81.02 | 155.18 |
| 80 | 80~20 | 30~20 | 150.56 | 60.00 | 210.55 |
| 70 | 70~20 | 30~20 | 153.21 | 59.96 | 213.16 |

Through the simulations with various battery capacities, the annual energy costs increased as the CD ranges decreased. For the scenario of CD range from 100% to 20%, the simulation results showed that the annual energy cost would be about 237.35 USD for 1.5 kWh battery capacity, about 188.60 USD for 2.0 kWh, and about 141.29 USD for 2.5 kWh. As the capacity of lithium ion batteries was increased from 1.5 kWh to 2.5 kWh, it was found that overall annual energy costs for various CD and CS ranges tended to decrease. Consequently the simulation showed that the more lithium ion battery capacity available, the more the annual energy cost savings can be achieved. In other words, expanding the CD range can be considered advantageous in terms of annual energy cost savings.

Unfortunately in Thailand, the situation is not that straightforward as the electrical infrastructure is not stable and thus, from time to time it may not be wise to fully count on electrical power. Therefore, more gasoline-heavy scenarios were investigated where the CD range was set at 100~20% while CS ranges were changed from 60~20%, 50~20%, 40~20%, 30~20%, 28~20%, 24~20%, and 22~20% as shown in Tables 5–7.

**Table 5.** Li-ion batt capacity 1.5 kWh result (CD range fixed).

| Initial SOC (%) | CD Range (%) | CS Range (%) | Annual Gasoline Cost (USD) | Annual Electricity Cost (USD) | Annual Energy Cost (USD) |
|---|---|---|---|---|---|
| 100 | 100~20 | 60~20 | 188.23 | 51.97 | 240.20 |
| 100 | 100~20 | 50~20 | 201.91 | 48.20 | 250.11 |
| 100 | 100~20 | 40~20 | 181.40 | 54.23 | 235.63 |
| 100 | 100~20 | 30~20 | 184.35 | 53.00 | 237.35 |
| 100 | 100~20 | 28~20 | 185.41 | 53.01 | 238.42 |
| 100 | 100~20 | 26~20 | 184.09 | 53.65 | 238.37 |
| 100 | 100~20 | 24~20 | 175.49 | 56.15 | 231.63 |
| 100 | 100~20 | 22~20 | 173.24 | 56.78 | 230.02 |

**Table 6.** Li-ion batt capacity 2.0 kWh result (CD range fixed).

| Initial SOC (%) | CD Range (%) | CS Range (%) | Annual Gasoline Cost (USD) | Annual Electricity Cost (USD) | Annual Energy Cost (USD) |
|---|---|---|---|---|---|
| 100 | 100~20 | 60~20 | 200.41 | 45.47 | 245.88 |
| 100 | 100~20 | 50~20 | 183.29 | 50.22 | 233.51 |
| 100 | 100~20 | 40~20 | 119.33 | 68.58 | 187.86 |
| 100 | 100~20 | 30~20 | 120.30 | 68.30 | 188.60 |
| 100 | 100~20 | 28~20 | 97.36 | 74.91 | 172.27 |
| 100 | 100~20 | 26~20 | 101.73 | 73.79 | 175.52 |
| 100 | 100~20 | 24~20 | 98.68 | 74.49 | 173.18 |
| 100 | 100~20 | 22~20 | 94.01 | 75.56 | 169.57 |

**Table 7.** Li-ion batt capacity 2.5 kWh result (CD range fixed).

| Initial SOC (%) | CD Range (%) | CS Range (%) | Annual Gasoline Cost (USD) | Annual Electricity Cost (USD) | Annual Energy Cost (USD) |
|---|---|---|---|---|---|
| 100 | 100~20 | 60~20 | 56.47 | 84.83 | 141.29 |
| 100 | 100~20 | 50~20 | 56.47 | 84.83 | 141.29 |
| 100 | 100~20 | 40~20 | 56.47 | 84.83 | 141.29 |
| 100 | 100~20 | 30~20 | 56.47 | 84.83 | 141.29 |
| 100 | 100~20 | 28~20 | 56.47 | 84.89 | 141.09 |
| 100 | 100~20 | 26~20 | 43.06 | 88.66 | 131.71 |
| 100 | 100~20 | 24~20 | 25.32 | 93.82 | 119.14 |
| 100 | 100~20 | 22~20 | 28.81 | 92.77 | 121.58 |

When the lithium ion battery capacity was 1.5 kWh and 2.0 kWh, the optimal CS range was 22~20% and the annual energy costs were 230.02 USD and 169.57 USD, respectively. When the lithium

ion battery capacity was 2.5 kWh, the optimal range was 24~20% and the annual energy cost was estimated to be 119.14 USD, which is excellent.

For Thailand, the results of the total annual cost of the ICE two-wheel vehicle and that of the plug-in series hybrid electric two-wheel vehicle over the years are plotted in the graph as shown in Figure 6. The graph showed that the recuperation of the initial investment for the plug-in series hybrid two-wheeler could take place in around 1.2 years with the fuel savings from the use of a battery. While further refinement for better analysis can be made, the initial results clearly showed that the conversion to a plug-in series hybrid electric two-wheel vehicle from a conventional ICE two-wheel vehicle would be advantageous in terms of annual cost.

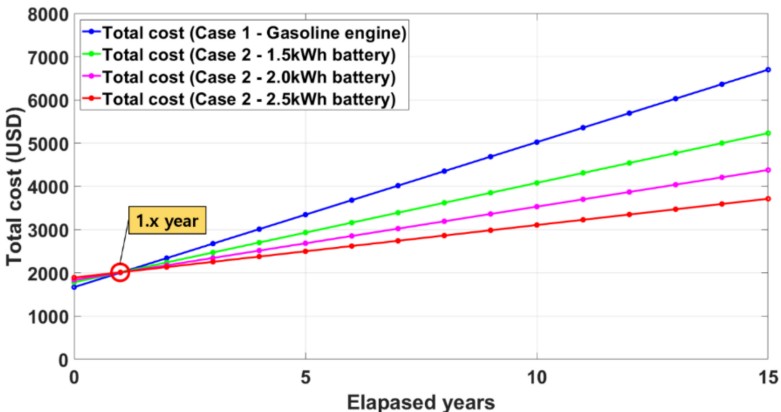

**Figure 6.** Total cost vs elapsed years (Thailand—1.5~2.5 kWh).

## 6.2. The Philippines

For the Philippines, the daily fuel cost of 0.80 USD was estimated from the simulation of the ICE two-wheel vehicle, thus the annual fuel cost was 290.44 USD. The Philippines has a better situation in terms of electrical infrastructure for regular households, and therefore the CD range was set more aggressively at 100~15% to be able to use more electricity. The simulation was able to predict that expanding the CD range would be advantageous in terms of annual energy costs, as the cost of gasoline in the Philippines is still quite expensive compared to that of electricity at home. To compare results to those of Thailand, simulations with battery capacities of 1.5 kWh, 2.0 kWh, and 2.5 kWh were performed respectively for various CS ranges from 55~15% down to 17~15% while CD range was fixed at 100~15%. For brevity, detailed tables are not included in the paper, but the results clearly demonstrated the advantage of the increased battery capacity and the extended CD range, which allowed the motor to use more electrical energy from the battery. For the battery capacities of 1.5 kWh and 2.0 kWh, the optimal CD and CS ranges were about the same, around CS 16~15%, with annual energy costs of 237.36 USD and 199.12 USD, respectively. For the battery capacity of 2.5 kWh, additional simulations were carried out in the CS range of 16.5~15% and 15.5~15% and the optimal CS range was found to be 16~15%, with the CD range of 100~15% resulting in an annual energy cost of 168.94 USD. Simulation results showed that increasing battery capacity from 1.5 kWh to 2.5 kWh could save about 33.56 USD on average.

For the Philippines, the results of the total annual cost of the ICE two-wheel vehicle and that of the plug-in series hybrid electric two-wheel vehicle over the years are plotted in the graph as shown in Figure 7. The graph showed that the recuperation of the initial investment for the plug-in series hybrid two-wheeler could occur in the vicinity of two years with the fuel savings from the use of battery. The main reason for the increase of recuperation period is the higher cost of electricity compared to that of Thailand. However, with the stable electrical infrastructures in the Philippines, it is still very favorable to convert to the plug-in series hybrid two-wheeler as the initial investment can be recuperated within two to three years depending on the aggressiveness of the operating strategy for

the series hybrid configuration. Nonetheless, the analysis can be further refined based on the current study and it is fully expected that we will continue the research in the near future.

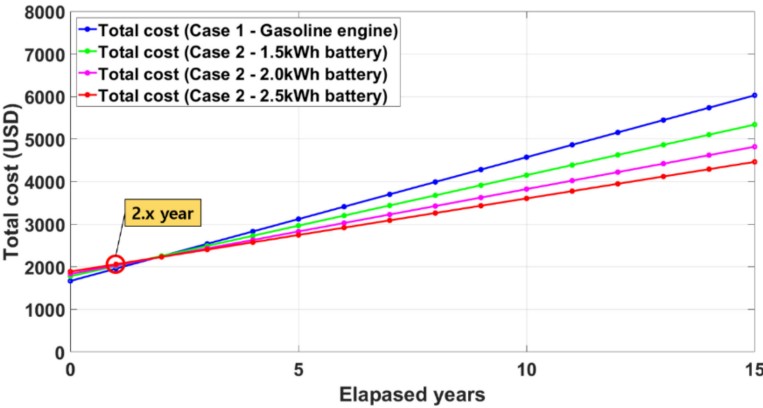

**Figure 7.** Total cost vs elapsed years (the Philippines—1.5~2.5 kWh).

## 7. Concluding Remarks

In the current study, forward energy simulation tools for a conventional ICE-based two-wheel vehicle and a plug-in series hybrid electric vehicle were developed and validated against the well-known energy balance tool from Argonne National Lab. Estimation of the recuperation period for the initial investment in the hybrid vehicle, and various configurations and scenarios were simulated with the in-house developed tools for four Southeast Asian countries where a significant portion of transportation relies on two-wheelers. Energy consumption from a variety of configurations and driving scenarios was collected with the actual cost-of-energy data from those countries to come up with a head-to-head fair market value comparison. In short, it is definitely favorable to move to the electrically supported hybrid configuration considering many aspects of the realistic scenarios discussed in the current paper. In the near future, further studies with more realistic maintenance costs and depreciation information will be conducted. With solid expectation of lower battery pack costs in coming years, it is clear the move to eco-friendly plug-in series hybrid electric vehicles will make life more pleasant on our planet earth and be more economically rewarding, especially in major cities of great Southeast Asian countries.

**Author Contributions:** W.C. is responsible for conceptualization, formulation, analysis and final preparation of the paper and S.Y. contributed in data and graph preparation. All authors have read and agreed to the published version of the manuscript.

**Funding:** This research received no external funding.

**Conflicts of Interest:** The authors declare no conflict of interest.

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
