# Peer review of "A Comparative Study of Initial Cost Recuperation Period of Plug-In Series Hybrid Electric Two-Wheel Vehicles in Southeast Asian Countries"

_sustainability, doi:10.3390/su122410340_

Round 1

Reviewer 1 Report

The paper is well presented and organized.

However, the topic is slightly novel and might be interesting for a limited number of readers.

Author Response

Please find the attached response file for you.

I really appreciate your effort for quick review.

Thanks a lot.

Reviewer 2 Report

The paper is an interesting study on a plug-in series hybrid electric two-wheel vehicles. However, there are a few major issues that need improvement:

  1. Literature research should be expanded. Ten references (only the two used in the introduction) are not enough to describe the current state of knowledge.
  2. If not necessary, abbreviations should not be used in the abstract, e.g. IC (lines 19 and 21).
  3. Line 49 – the two dots at the end of the sentence.
  4. Line 5 – conjunctions in section titles should be written in lowercase.
  5. Lines: 59–60, 87, 109, 146, 237, 251 – section titles should not be entirely capitalized.
  6. Lines: 89, 90 – Abbreviations like NYCC (however known) should be explained.
  7. Lines: 122–123 – the statement 'The total annual fuel cost was estimated simple by adding the initial two-wheel vehicle price to the annual fuel cost for all cases.' is not clear. Why is the vehicle price part of the annual fuel cost? I don't see a reflection of this in Tables 2–7. Maybe it was about 'the total annual maintenance cost'? 
  8. Section 7:
    1. please standardize the currency (formerly Won, here USD is used),
    2. the relationship between section 7 and the rest of the manuscript is poorly described; what part of the vehicle price is the price of the battery, what is the expected dynamics of the decline in battery prices and its impact on the vehicle price; additional calculations would be advisable.

To sum up, an interesting topic, but major changes are required.

Author Response

Response to the comments from Reviewer 2

First of all, I really appreciate careful review from Reviewer 2. Based on the reviewer’s comments, I have revised accordingly as follows.

Comment 1

Literature research should be expanded. Ten references (only the two used in the introduction) are not enough to describe the current state of knowledge.

Response 1

I have researched a few more reference materials and incorporated them in the introduction section. In the revised manuscript, blue color is used to highlight the changes.

Comment 2

If not necessary, abbreviations should not be used in the abstract, e.g. IC (lines 19 and 21).

Response 2

I have fixed based on the recommendation. Thank you very much.

Comment 3

Line 49 – the two dots at the end of the sentence.

Response 3

I have corrected the mistake based on the recommendation

Comment 4

Line 5 – conjunctions in section titles should be written in lower case.

Response 4

I have corrected the titles based on the recommendation. Thank you very much for pointing out the details.

Comment 5

Lines: 59–60, 87, 109, 146, 237, 251 – section titles should not be entirely capitalized.

Response 5

I have corrected the titles based on the recommendation. Thank you very much.

Comment 6

Lines: 89, 90 – Abbreviations like NYCC (however known) should be explained.

Response 6

I have added a full name for NYCC which is the “New York City Cycle”. Thanks a lot.

Comment 7

Lines: 122–123 – the statement 'The total annual fuel cost was estimated simple by adding the initial two-wheel vehicle price to the annual fuel cost for all cases.' is not clear. Why is the vehicle price part of the annual fuel cost? I don't see a reflection of this in Tables 2–7. Maybe it was about 'the total annual maintenance cost'?

Response 7

You are correct. What I meant was the total maintenance cost. Thanks for correcting the mistake.

Comment 8

Please standardize the currency (formerly Won, here USD is used)

Response 8

Based on the Reviewer’s comment, I changed them all in USD so that the reader could figure out the values more conveniently. I really appreciate the reviewer’s careful work.

Comment 9

The relationship between section 7 and the rest of the manuscript is poorly described; what part of the vehicle price is the price of the battery, what is the expected dynamics of the decline in battery prices and its impact on the vehicle price; additional calculations would be advisable.

Response 9

As commented by the Reviewer, I would agree that the relationship between Section 7 and the rest was not direct. All I wanted to point out was that the initial purchase cost would decrease in the future. Since the main focus of the paper is to understand the benefit of hybrid two wheeler over the time, I have decided to Section 7. I appreciate your recommendation and review efforts.

Comment 10

To sum up, an interesting topic, but major changes are required.

Response 10

Thank you very much for your interests and support. I really appreciate your help.

Reviewer 3 Report

This research focused on the operating cost of two-wheel plug-in hybrid electric vehicles. This article is very interesting to read.

In this article, authors mentioned that the two-wheel model was validated against the result of Autonomie from Argonne. Authors should provide more simulation information about the validation to show the model was reliable.

With ICE and BLDC motor as the main power sources, authors should include more detail descriptions about these two models applied in the simulation. Authors should also provide the efficiency data of these two during simulation.

For initial cost, authors only mentioned the values were obtained through data survey at the Alibaba site. Authors should include the survey result and cost number of two-wheel vehicles applied in this article.

The battery cost changed a lot during the pass few years. Authors should provide the battery price applied in the article. Especially, the battery would decay in a few years, and it would require a replacement. This should also be included in the operating cost.

Author Response

Response to the comments from Reviewer 3

It was my honor to hear such a careful review from Reviewer 3. I really appreciate the efforts of Reviewer 3. Based on the reviewer’s comments, I have revised the manuscript accordingly and as much as possible.

Comment 1

In this article, authors mentioned that the two-wheel model was validated against the result of Autonomie from Argonne. Authors should provide more simulation information about the validation to show the model was reliable. With ICE and BLDC motor as the main power sources, authors should include more detail descriptions about these two models applied in the simulation. Authors should also provide the efficiency data of these two during simulation.

Response 1

I have revised Section 4 to accommodate the comment from Reviewer 3. I have provided a little more information about the simulation with the exemplary 3.0 kW BLDC motor torque-RPM-efficiency map and the 125CC engine torque-RPM-efficiency-fuel consumption map, respectively. I really appreciate the review efforts from Reviewer 3.

Comment 2

For initial cost, authors only mentioned the values were obtained through data survey at the Alibaba site. Authors should include the survey result and cost number of two-wheel vehicles applied in this article.

Response 2

During the course of our study, actual market price survey was not conducted due to resource limitation. However, the information from the website were known to be quite sensitive to the actual market trend and therefore, we have decided to use the data from the web site. For the next phase study, we will try to run a market survey as much as we can. I would surely keep the comment from Reviewer 3 in mind for future analysis.

Comment 3

The battery cost changed a lot during the pass few years. Authors should provide the battery price applied in the article. Especially, the battery would decay in a few years, and it would require a replacement. This should also be included in the operating cost.

Reviewer 3

For the current study, the battery prices were quoted at early 2019. I would fully agree with the comment from Reviewer 3 about the depreciation of the battery quality and the possible replacement cost in the future. However, during the early years of ownership, the results showed the benefit of the battery and the typical life of battery would last over 7~10 years nowadays so I assumed the replacement cost of the battery would be reasonably similar to the cost of engine replacement and maintenance cost. I would really appreciate if Reviewer 3 could agree with my idea as an initial report about the study of initial cost recuperation period among different countries.

Reviewer 4 Report

This manuscript presents an interesting study on the feasibility of plug in hybrid electric two wheelers for the Southeast Asian market. Whilst the concept is interesting it is not possible to assess the accuracy of the study since few details with respect to the model are presented. In a more general perspective, I also feel that the extra complexity of a plug-in hybrid is not suited to low power two-wheelers and it would be better to move directly to electric, even with grid insecurities.

Requested changes:

Could you please provide some more detail on how different aspects of the model were simulated, how was the engine, motor and battery modelled. How was the model validated and how accurate to you perceive it to be?

The purchase prices are not included in the report, what are the purchase prices of the ICE and PHEV vehicles? I am not aware of any commercial PHEV motorcycles.

Please include a fully electric motorcycle in your comparison. I understand the grid limitations, but you mention over 90% grid penetration in these countries and it is important to show the full range from ICE to full EV.

There is not really a sufficient review of the literature, please add some relevant studies on cost analysis of electric vehicles.

Abstact line 15: It is not obvious in reality since electric grid systems for 15 stable supply of electricity are not there yet. Does not make sense, is this refereeing to fully electric two-wheelers?

Please make fig 1 a bit easier to read with larger text, possibly placing the vehicles side by side to match the 1 column journal format.

No literature review

Define BLDC and other acronyms during first use.

Figure 3, subsystem text is too small to read

It is useful to use the same currency across the different countries, but I feel it would appeal more to the international readership if it were expressed in US dollars or Euros.

The split between CD range and CS range is a little difficult to follow, could you please include a figure showing how the SOC changes with distance for a certain initial SOC, CD range and CS range?

The final column in tables 2, 3 and 4 etc should be Annual energy cost, not annual fuel cost as electricity is not a fuel.

Table 7 – Does the model always finish on the same battery SOC? With wider CS range the increased cost may be due to the vehicle finishing the test on a higher SOC.

Figure 4 – Y axis says maintenance, this should be total cost, remove title as this information is in the caption and annotation says 1.x years.

Do you have permission to use figure 6?

Author Response

Response to the comments from Reviewer 4

It was so nice to get comments from such a careful reviewer as Reviewer 4. I truly appreciate the comments from Reviewer 4. Accordingly, I have explained the situation and revised the manuscript as recommended as much as possible.

Comment 1

This manuscript presents an interesting study on the feasibility of plug in hybrid electric two wheelers for the Southeast Asian market. Whilst the concept is interesting it is not possible to assess the accuracy of the study since few details with respect to the model are presented. In a more general perspective, I also feel that the extra complexity of a plug-in hybrid is not suited to low power two-wheelers and it would be better to move directly to electric, even with grid insecurities.

Response 1

I have revised Section 4 to accommodate the comment from Reviewer 4. I have provided a little more information about the simulation with the exemplary 3.0 kW BLDC motor torque-RPM-efficiency map and the 125CC engine torque-RPM-efficiency-fuel consumption map, respectively. I really appreciate the review efforts of Reviewer 4. In regards to the EV adoption instead of PHEV adoption, I had a similar question and consulted with local experts in the field of vehicle electrification. After the consultation about realistic situation of each country for EV and PHEV scenario, I was convinced by them that PHEV would be more dependable for majority of people who would use the motorcycle for their day-to-day life. Again, the current study was conducted based on the guide and consultation by the local specialist and I would appreciate if Reviewer 4 could understand the current circumstances and allow us to be able to submit the paper in the journal.

Comment 2

Could you please provide some more detail on how different aspects of the model were simulated, how was the engine, motor and battery modelled. How was the model validated and how accurate to you perceive it to be?

Response 2

I appreciate the comment from Reviewer 4. I have revised Section 4 to accommodate the comment from Reviewer 4. I have provided a little more information about the simulation with the exemplary 3.0 kW BLDC motor torque-RPM-efficiency map and the 125CC engine torque-RPM-efficiency-fuel consumption map, respectively. I have also included a more description how I validated the in-house model. I really appreciate the review efforts from Reviewer 4.

Comment 3

The purchase prices are not included in the report, what are the purchase prices of the ICE and PHEV vehicles? I am not aware of any commercial PHEV motorcycles.

Response 3

I have revised the paragraph in Section 5 as follows to answer the question from Reviewer 4.

The initial purchase cost of a typically available ICE two-wheel vehicle was obtained through data survey at the Alibaba internet shopping mall and the cost of a plug-in series hybrid electric two-wheel vehicle was estimated by simple adding the motor, controller and battery cost. In this estimation, cost of the engineering development was not considered in order to compare the products virtually in mass production accordingly.

I truly appreciate the efforts by Reviewer 4 to make things clear.

Comment 4

Please include a fully electric motorcycle in your comparison. I understand the grid limitations, but you mention over 90% grid penetration in these countries and it is important to show the full range from ICE to full EV.

Response 4

While I fully understand and value the suggestion from Reviewer 4, I have conducted the current study based on the local researchers Thailand and Philippine as I mentioned in Response 1. Therefore, I have the results for ICE and PHEV configurations. I would like to follow the suggestion from Reviewer 4 for the next study. But due to the time and resource limitation for the current version of paper, I would really like to ask a great favor from Reviewer 4 to allow the publication to move as is without further delay.

Comment 5

There is not really a sufficient review of the literature, please add some relevant studies on cost analysis of electric vehicles.

Response 5

I have researched for more reference materials and added many mores in the revised manuscript.

Thank you very much for the comment.

Comment 6

Abstract line 15: It is not obvious in reality since electric grid systems for stable supply of electricity are not there yet. Does not make sense, is this refereeing to fully electric two-wheelers?

Response 6

What I was trying to describe is “consistent and stable” and this is also revised in the abstract. As Reviewer 4 mentioned, electric supply was not that consistent to use a pure electric vehicle as a solely reliable device. Therefore, there are a lot of demands for PHEV and thus the current study was suggested and conducted.

Thanks a lot for the clarification and I really appreciated that.

Comment 7

Please make fig 1 a bit easier to read with larger text, possibly placing the vehicles side by side to match the 1 column journal format.

Response 7

Based on the comment from Reviewer 4, I have modified Fig. 1 accordingly.

Comment 8

No literature review

Response 8

I have done more literature search and revised the manuscript to include them in response to the comment from Reviewer 4

I appreciated your suggestion. Thanks a lot.

Comment 9

Define BLDC and other acronyms during first use.

Response 9

I have revised the manuscript to reflect the comment from Reviewer 4. I have taken care of a few more acronyms during the first use of them as well.

Thanks for the comments.

Comment 10

Figure 3, subsystem text is too small to read

Response 10

I have replaced the figure with modification as pointed out. Thanks a lot.

Comment 11

It is useful to use the same currency across the different countries, but I feel it would appeal more to the international readership if it were expressed in US dollars or Euros.

Response 11

I do agree with the comment from Reviewer 4 for this matter. I have reflected the changes in the manuscript everywhere needed. I have also upgraded the figures to reflect these changes.

I really appreciated the efforts from Reviewer 4. Thank you very much.

Comment 12

The split between CD range and CS range is a little difficult to follow, could you please include a figure showing how the SOC changes with distance for a certain initial SOC, CD range and CS range?

Response 12

In the manuscript of Section 6.1, I have described the details of the simulation scenario as follows.

For example, the scenario of 90% ~ 20% of CD range and 30% ~ 20% of CS range means that the electricity from the battery SOC 90% down to 20% is used first and then, the ICE generator kicks in to produce electricity until the SOC reaches up to 30%. Once, the SOC reaches to 30% SOC, then the ICE stops and the traction motor simply uses the electricity again from 30% SOC down to 20% SOC. This process repeats until the total driving distance of 39.2 km/day is achieved.

I really appreciated the comment from Reviewer 4.

Comment 13

The final column in tables 2, 3 and 4 etc should be Annual energy cost, not annual fuel cost as electricity is not a fuel.

Response 13

I really appreciated this comment since it cleared up many ambiguous points. I have modified the manuscript in all tables.

Comment 14

Table 7 – Does the model always finish on the same battery SOC? With wider CS range the increased cost may be due to the vehicle finishing the test on a higher SOC.

Response 14

The simulation stopped as it completed the driving distance. So, it might have finished within the middle of the CS range of SOC. This is also described in Section 6.1. It might have caused some uncertainty. However, the driving distance was the main factor when it came to the comparative study, the results were thought to be immune to the CS range.

Thanks a lot for pointing this to make it clear.

Comment 15

Figure 4 – Y axis says maintenance, this should be total cost, remove title as this information is in the caption and annotation says 1.x years.

Response 15

Based on the comment from Reviewer 4, I have revised the figure accordingly.

Thanks a lot.

Comment 16

Do you have permission to use figure 6?

Response 16

A similar comment was also made by other reviewers. After the careful review of the current manuscript, I have realized the I could delete Section 7 without any impact on the current study. So, I have completely deleted the old Section 7.

Round 2

Reviewer 2 Report

I am pleased to write that after improvements, I may suggest publishing the manuscript.

Author Response

Response to the comments from Reviewer 2

First of all, I really appreciate careful review from Reviewer 2. Based on the reviewer’s comments, I have revised accordingly as follows.

Comment 1

I am pleased to write that after improvements, I may suggest publishing the manuscript.

Response 1

I DO appreciate the thoughtful review from Reviewer 2. I would appreciate further if you could sign your review report in the Review Report Form.

Reviewer 3 Report

This research provided the operating cost of two-wheel plug-in hybrid electric vehicles. This This article is interesting to read.

Author Response

Response to the comments from Reviewer 3

It was my honor to hear such a careful review from Reviewer 3. I really appreciate the efforts of Reviewer 3. Based on the reviewer’s comments, I have revised the manuscript accordingly and as much as possible.

Comment 1

This research provided the operating cost of two-wheel plug-in hybrid electric vehicles. This article is interesting to read.

Response 1

I DO appreciate the thoughtful review from Reviewer 3. Thank you very much.

Reviewer 4 Report

The changes made by the authors have improved the quality of the paper. I do think a few of the points could have been better addressed and would like to see the following changes made:

Either a fully electric motorcycle variant in the model or acknowledgement that a fully electric two-wheeler could give a lower overall operating cost but is beyond the scope of the study.

The term ‘maintenance’ is used incorrectly. My understanding of maintenance refers to the upkeep and servicing of the vehicle, including replacement parts and not the fuel costs. In figures 6 and 7 it should be total cost on the y-axis as this cost is not either annually occurring or for maintenance.

Graphs that have captions should not have titles as the information should be in the caption, for example in figures 4, 5, 6 and 7.

Author Response

Response to the comments from Reviewer 4

I DO appreciate the comments from Reviewer 4 and I have revised the manuscript accordingly.

General Note from Reviewer 4

The changes made by the authors have improved the quality of the paper. I do think a few of the points could have been better addressed and would like to see the following changes made:

Comment 1

Either a fully electric motorcycle variant in the model or acknowledgement that a fully electric two-wheeler could give a lower overall operating cost but is beyond the scope of the study.

Response 1

I fully agree with this comment and I have made an additional description follows (Line 86 ~ 89)

Nevertheless, a fully electric two-wheeler could give a lower overall operating cost but was beyond the scope of the current study as the research focus was on the plug-in hybrid electric two-wheeler under the influence of rather unstable power grid situation.

Comment 2

The term ‘maintenance’ is used incorrectly. My understanding of maintenance refers to the upkeep and servicing of the vehicle, including replacement parts and not the fuel costs. In figures 6 and 7 it should be total cost on the y-axis as this cost is not either annually occurring or for maintenance.

Response 2

Following the comment from Reviewer 4, I have revised the graphs accordingly. Thank you very much to make them clear. I have also taken care of “maintenance” in the manuscript to accommodate the comment from Reviewer 4.

Comment 3

Graphs that have captions should not have titles as the information should be in the caption, for example in figures 4, 5, 6 and 7.

Response 3

I agreed to the comment and I have revised the plots accordingly.

Thank you very much for your careful advice.
